# Mouse Model of Parkinson’s Disease with Bilateral Dorsal Striatum Lesion with 6-Hydroxydopamine Exhibits Cognitive Apathy-like Behavior

**DOI:** 10.3390/ijms25147993

**Published:** 2024-07-22

**Authors:** Masato Okitsu, Masayo Fujita, Yuki Moriya, Hiroko Kotajima-Murakami, Soichiro Ide, Rika Kojima, Kazunari Sekiyama, Kazushi Takahashi, Kazutaka Ikeda

**Affiliations:** 1Addictive Substance Project, Tokyo Metropolitan Institute of Medical Science, Tokyo 156-8506, Japan; okitsu-ms@igakuken.or.jp (M.O.); fujita-ms@igakuken.or.jp (M.F.); moriya-yk@igakuken.or.jp (Y.M.); kotajima-hr@igakuken.or.jp (H.K.-M.); ide-si@igakuken.or.jp (S.I.); 2Department of Neurology, Tokyo Metropolitan Neurological Hospital, Tokyo 183-0042, Japan; kazushi_takahashi@tmhp.jp; 3Laboratory of Molecular Pathology and Histology, Tokyo Metropolitan Institute of Medical Science, Tokyo 156-8506, Japan; kojima-rk@igakuken.or.jp (R.K.); sekiyama-kz@igakuken.or.jp (K.S.); 4Department of Neuropsychopharmacology, National Institute of Mental Health, National Center of Neurology and Psychiatry, Tokyo 187-8553, Japan

**Keywords:** Parkinson’s disease, apathy, cognitive apathy, 6-hydroxydopamine, novelty seeking, social novelty seeking

## Abstract

Among the symptoms of Parkinson’s disease (PD), apathy comprises a set of behavioral, affective, and cognitive features that can be classified into several subtypes. However, the pathophysiology and brain regions that are involved in these different apathy subtypes are still poorly characterized. We examined which subtype of apathy is elicited in a mouse model of PD with 6-hydroxydopamine (6-OHDA) lesions and the behavioral symptoms that are exhibited. Male C57/BL6J mice were allocated to sham (n = 8) and 6-OHDA (n = 13) groups and locally injected with saline or 4 µg 6-OHDA bilaterally in the dorsal striatum. We then conducted motor performance tests and apathy-related behavioral experiments. We then pathologically evaluated tyrosine hydroxylase (TH) immunostaining. The 6-OHDA group exhibited significant impairments in motor function. In the behavioral tests of apathy, significant differences were observed between the sham and 6-OHDA groups in the hole-board test and novelty-suppressed feeding test. The 6-OHDA group exhibited impairments in inanimate novel object preference, whereas social preference was maintained in the three-chamber test. The number of TH+ pixels in the caudate putamen and substantia nigra compacta was significantly reduced in the 6-OHDA group. The present mouse model of PD predominantly showed dorsal striatum dopaminergic neuronal loss and a decrease in novelty seeking as a symptom that is related to the cognitive apathy component.

## 1. Introduction

Parkinson’s disease (PD) is a progressive neurodegenerative disorder that is characterized by four major motor symptoms: resting tremor, bradykinesia, rigidity, and loss of postural reflexes [1]. Nonmotor symptoms, such as various psychiatric symptoms and autonomic dysfunction, are also associated with PD, considerably contributing to a decline in quality of life in patients [2]. Neuronal loss in the substantia nigra, which causes striatal dopamine deficiency, and intracellular inclusions that contain aggregates of α-synuclein, are neuropathological hallmarks of PD [3].

Among nonmotor symptoms of PD, apathy consists of a set of behavioral, affective, and cognitive features. Apathy can be defined as a state of low motivation that manifests as a decrease in goal-directed behaviors. It can be variably characterized by a reduction in interests and blunted emotions that cannot be attributed to lower levels of consciousness, cognitive impairment, or emotional distress [4]. The comorbidity of apathy is not limited to advanced stages of PD. It can also be present from early stages of the disease, and its prevalence is reported to range from 16.5% to 60% of PD patients [5]. In PD patients, apathy-associated complications are important and play a key role in daily functioning. For example, apathy can lead to a poor response to motor symptom treatment and greater healthcare spending of both patients and their caregivers, and it can negatively influence general cognitive function and financial capacity performance [6,7]. Additionally, apathy was reported to be an independent nonmotor symptom that determined poor health-related quality of life in patients with early PD [8]. Although apathy causes such significant losses in the lives of patients, there is no specific evidence-based treatment [9].

Various symptoms of apathy can be classified into several subtypes. The responsible neural systems, comprising multiple anatomical sites, for each subtype are assumed to be different and partially interact [10]. Several classifications of apathy have been proposed in the literature, which are generally classified into “cognitive”, “emotional-affective (motivational)”, and “self(auto)-activation”, and there is a tendency to classify symptoms that are related to social interaction separately as “social” apathy [10,11,12,13,14]. Dysfunction of the mesocorticolimbic pathway leads to emotional-affective apathy, and dysfunction of the dorsolateral prefrontal cortex (PFC) and caudate nucleus leads to cognitive apathy [10]. Although social apathy has not been investigated sufficiently in previous work, the anterior cingulate cortex (ACC) and PFC have been suggested to play a role in processing social information and social interaction [15,16,17]. Each apathy subtype exhibits different kinds of symptoms. For example, disruption of the planning aspect of goal-directed behavior leads to cognitive apathy and clinically manifests as lower interest and lower curiosity [18]. Emotional-affective apathy exhibits a blunted emotional response when the patient is confronted with upsetting news or watches something humorous. For example, patients can exhibit little concern for the health and well-being of themselves or their family members [19]. In social apathy, a low level of engagement in social interactions (e.g., patients cannot initiate conversations without being prompted) may be observed [20].

Based on these considerations, a single agent is expected to not be effective for treating all symptoms of apathy. Effective treatments may differ depending on symptom subtypes. To develop therapies based on different types of symptoms, correspondence between apathy subtypes and symptoms needs to be investigated in experimental animals.

Experiments in animals are indispensable in the search for effective treatments for symptoms of apathy. Several animal studies have investigated psychiatric symptoms of PD, including apathy [21,22,23,24,25]. However, few animal studies have focused on behavior from the standpoint of apathy subtypes, including cognitive, emotional-affective, and social aspects of apathy [26,27,28]. 

In the present study, we generated a mouse model of PD with local 6-hydroxydopamine (6-OHDA) administration, which induces dopaminergic neuronal cell death, bilaterally in the dorsal striatum. Based on previous reports [10,12], we hypothesized that the corresponding neural network, including the dorsal striatum, would be disrupted and result in behavioral abnormalities that are attributable to the cognitive apathy subtype. We examined which subtype of apathy is actually observed by dopaminergic neuronal loss in the dorsal striatum and the behavioral symptoms that are elicited.

Anhedonia was originally conceptualized only as an inability to experience pleasure, but it is now recognized as including the loss of motivation to act in pursuit of pleasure. Although apathy and anhedonia represent different symptoms, recent studies suggest that they partly overlap and share neural mechanisms that are related to reward processing [29]. Therefore, anhedonia was also evaluated in the present study as an apathy-like behavior. 

The experimental concept and design are depicted in Figure 1A,B.

## 2. Results

### 2.1. Decrease in Tyrosine Hydroxylase (TH) Immunostaining in the Caudate Putamen (CPu) and Substantia Nigra Compacta (SNc) by 6-OHDA Injections

One mouse in the 6-OHDA group exhibited almost no TH-positive neuronal loss in the SNc, suggesting a failure of 6-OHDA lesioning. Therefore, this mouse was excluded from all evaluations. We analyzed the loss of TH-positive neurons in the CPu and nucleus accumbens (NAcc) as the cephalic slice and in the SNc and ventral tegmental area (VTA) as the caudal slice (Figure 2A). Histological analysis revealed the significant loss of TH-positive pixels (mean ± standard error of the mean [SEM]) in the CPu (sham group: 140.6 ± 6.8; 6-OHDA group: 97.3 ± 5.3; *p* = 0.000086; Figure 2B) and cell counts in the SNc (sham group: 111.6 ± 4.4; 6-OHDA group: 75.4 ± 2.2; *p* = 0.00000033; Figure 2C) in the 6-OHDA group compared with the sham group. In the NAcc and VTA, no significant reductions in pixels or cell counts were found in the 6-OHDA group compared with the sham group (NAcc, sham group: 30.6 ± 1.3; 6-OHDA group: 29.2 ± 1.2, *p* = 0.43; VTA, sham group: 91.6 ± 5.2; 6-OHDA group: 84.4 ± 2.0, *p* = 0.22; Figure 2D,E). These results indicate that TH neurons in the CPu and SNc, but not in the VTA or NAcc, were dominantly lesioned with 6-OHDA in our mouse model.

### 2.2. Motor Performance Decline in the 6-OHDA Group

To ensure validity of the mouse model as a model of PD, three motor performance tests (rotarod, pole, and balance beam) were conducted to confirm motor dysfunction. In the rotarod test, the sham group did not change significantly (before sham lesioning: 250.63 ± 19.96 s; after sham lesioning: 259.54 ± 22.58 s; *p* = 0.84), but the 6-OHDA group exhibited impairments after lesioning (before lesioning: 224.25 ± 14.26 s; after lesioning: 146.97 ± 14.39 s; *p* = 0.0015). After lesioning, a comparison between the sham group and 6-OHDA group showed a significant decrease in motor performance in the 6-OHDA group (*p* = 0.0011; Figure 3A). In the pole test, the time required for the mice to orient themselves in a downward direction, T_turn_, did not indicate impairments in either group before or after lesioning (sham group: before sham lesioning: 2.65 ± 0.32 s, after sham lesioning: 2.29 ± 0.16 s, *p* = 0.38; 6-OHDA group: before lesioning: 2.61 ± 0.28 s, after lesioning: 3.15 ± 0.39 s, *p* = 0.47). A comparison between the sham group and 6-OHDA group showed no significant impairments in the 6-OHDA group (*p* = 0.062). For the time required for the mice to descend to the base of the pole, T_down_, in the pole test, neither group changed significantly (sham group: before sham lesioning: 9.26 ± 0.80 s, after sham lesioning: 8.48 ± 0.45 s, *p* = 0.38; 6-OHDA group: before lesioning: 9.39 ± 0.51 s, after lesioning: 10.28 ± 0.37 s, *p* = 0.15). After lesioning, a comparison of the sham group and 6-OHDA group showed a significant increase in time in the 6-OHDA group (*p* = 0.0067; Figure 3B). In the balance beam test, there was no significant change in the sham group (before sham lesioning: 9.33 ± 0.78 s, after sham lesioning: 8.42 ± 0.63 s; *p* = 0.313). A longitudinal comparison within the 6-OHDA group showed a decline in times, although this was not significant (before lesioning: 8.43 ± 0.31 s; after lesioning: 11.32 ± 1.08 s; *p* = 0.051). A postoperative comparison of the sham group and 6-OHDA group showed a significant increase in time in the 6-OHDA group (*p* = 0.025; Figure 3C). The 6-OHDA group exhibited a decline in most of the motor performance tests, indicating that they could be used as a PD model.

### 2.3. Anhedonia-like Behavior Suggested by the Novelty-Suppressed Feeding Test (NSFT) but Not by the Sucrose Preference Test (SPT) in the 6-OHDA Group

The sucrose preference test (SPT) is a behavioral test that assesses anhedonia, and various protocols exist [30,31,32]. Rodents have an innate preference for palatable, sweet solutions. This preference is thought to correlate with the pleasant sensation that is experienced by animals when they ingest sucrose. Therefore, lower sucrose preference is interpreted as a state of lower pleasure or anhedonia [33]. The novelty-suppressed feeding test (NSFT) is used to assess anxiety [34] but is also used to assess a component of anhedonia [35,36].

In the SPT, the sucrose preference index (percentage) was not different between the sham and 6-OHDA groups (sham group: 61.94% ± 7.17%, 6-OHDA group: 64.92% ± 3.24%; *p* = 0.683; Figure 4A). In the NSFT, one mouse in the sham group and three mice in the 6-OHDA group reached the 6 min cutoff time before eating the food pellet. The mean latency to reach the pellet and take a bite with the forelimbs was significantly longer in the 6-OHDA group (sham group: 159.09 ± 32.42 s; 6-OHDA group: 255.63 ± 29.45 s; *p* = 0.038; Figure 4B). The result of the SPT did not suggest anhedonia, but the NSFT did.

### 2.4. Anxiety Was Not Observed but Novelty-Seeking Behavior Decreased in the 6-OHDA Group

The open-field test (OFT) can be used as a test of anxiety because as an anxiety-like behavior, mice do not prefer entering the central area of a test space [37]. In the OFT, the number of times each mouse entered the four center square zone was not different between the sham and 6-OHDA groups (sham group: 33.13 ± 4.56; 6-OHDA group: 29.91 ± 2.88; *p* = 0.54; Figure 5A). The time each mouse was in the four center square zone was also not different (sham group: 56.74 ± 8.04 s; 6-OHDA group: 54.51 ± 9.52 s; *p* = 0.867; Figure 5A). There was no difference in the total distance traveled between the sham and 6-OHDA groups (sham group: 24.00 ± 2.00 m; 6-OHDA group: 24.66 ± 2.07 m; *p* = 0.827; Figure 5A), suggesting that spontaneous locomotor activity was not affected by 6-OHDA. The hole-board test (HBT) is a behavioral test that evaluates novelty-seeking behavior [38,39]. A decrease in the number of head-dipping behaviors in the novel environment within the time limit suggests a decrease in innate novelty-seeking behavior in rodents. In the HBT, the number of times the mice introduced their head in the holes significantly decreased in the 6-OHDA group compared with the sham group (sham group: 16.63 ± 1.94; 6-OHDA group: 10.09 ± 1.59; *p* = 0.018; Figure 5B). Furthermore, statistical analysis of the correlation between the HBT results and pathological analysis results in the 6-OHDA group using the Pearson correlation coefficient showed a significant positive correlation with SNc cell counts (correlation coefficient = 0.694, 95% confidence interval: 0.162–0.914, *p* = 0.0177). These results indicated a significant decrease in novelty-seeking behavior but not anxiety-like behavior.

### 2.5. Self-Care Behavior Was Maintained in the 6-OHDA Group

As a component of apathy-like symptoms, self-care activity was assessed in the nest-building test (NBT) and splash test (ST). The NBT can be used to examine apathy symptoms [40], although a wide variety of causes, such as general condition, distress, and pain, may also provoke deficits in nesting behavior [41]. The ST is used to examine apathy symptoms and self-care behavior [42]. In the NBT, the nest-building score decreased in the 6-OHDA group (1.45 ± 0.17) compared with the sham group (2.13 ± 0.30), but this difference was not statistically significant (*p* = 0.052; Figure 6A). In the ST, the latency to the first grooming bout was longer in the 6-OHDA group (6.23 ± 0.37 s) compared with the sham group (2.99 ± 1.48 s), but this difference was also not statistically significant (*p* = 0.074), and the SDs were large. The total number of grooming behaviors was also not significantly different between the sham and 6-OHDA groups (sham group: 21.88 ± 3.88; 6-OHDA group: 24.20 ± 4.50; *p* = 0.709; Figure 6B). In the NBT, a nonsignificant tendency toward a decrease in nesting activity was observed. No apparent decline in grooming behavior was indicated in the ST. In summary, these results did not suggest an obvious impairment in self-care behavior.

### 2.6. Object/Inanimate Novelty Was Impaired but Social Novelty Was Maintained in the 6-OHDA Group

As social interaction was assessed in the three-chamber test. The first session (session 1) assessed preference for a social object (mouse) compared with a novel, nonanimal object. Another session (session 2) assessed preference for a newer individual rather than a familiar one [43]. The time that the head of the mouse was within the zone 30 mm around the holding cage (interaction time) was assessed. In session 1, assessment of sociability, the interaction time for the novel object significantly decreased in the 6-OHDA group compared with the sham group (sham group: 111.99 ± 5.48 s; 6-OHDA group: 89.98 ± 7.39 s; *p* = 0.0397; Figure 7A). There was no significant difference in interaction time for the social object between the two groups (sham group: 115.86 ± 7.70 s; 6-OHDA group: 122.84 ± 8.75 s; *p* = 0.575; Figure 7A). These results indicate that social behavior was maintained in both groups of mice. However, the exploration time for a novel object significantly decreased in the 6-OHDA group. In session 2, the social novelty preference test, the interaction time for the familiar mouse and stranger mouse did not significantly differ between the two groups (familiar mouse: sham group: 59.68 ± 6.98 s, 6-OHDA group: 67.95± 10.13 s, *p* = 0.543; stranger mouse: sham group: 99.14 ± 13.60 s, 6-OHDA group: 94.25 ± 8.68 s, *p* = 0.755; Figure 7B). These results indicate that social novelty seeking was maintained.

## 3. Discussion

In the present study, we adopted 6-OHDA lesioning methods that have been used extensively in previous studies as a mouse model of PD, including studies of nonmotor symptoms [44,45]. Although negative opinions have begun to be advocated in recent years, male mice have been utilized to avoid confounding effects of the female estrous cycle and other potential variables that may interfere with experimental outcomes [46,47]. Following this conventional practice, we used only male mice in the present experiments. The validity of the PD model was confirmed by motor tests and pathological evaluations. Behavioral tests of apathy-like behavior were then conducted. Significant differences were found in the HBT and three-chamber test, suggesting a reduction in object/inanimate novelty seeking in the 6-OHDA group. Significant differences were also found in the NSFT. The present study is novel and valuable in that animal behavioral experiments showed that reduced novelty-seeking behavior seems to be a symptom with a strong cognitive apathy component among the apathy symptoms of PD. Furthermore, it is interesting to note that while object/inanimate novelty seeking decreased, social novelty was maintained.

The administration of 6-OHDA reduced dopaminergic neurons in the CPu and SNc to 69% and 68% of the sham group, respectively. The percentage of dopamine neuronal loss in the striatum was reported to be 63–74% in a previous study [48]. The degree of dopaminergic neuronal loss in the present study was at the same level and is considered to correspond to the early stage of PD. Motor performance significantly decreased in our model, indicating that the PD mouse model may mimic an earlier stage of PD. In the pole test, T_turn_ measures the time required for the mouse to change head direction, and T_down_ measures the time to descend the pole. The present results showed significant differences only in T_down_ and not in T_turn_, which may indicate mild motor symptoms in the present mouse model.

Apathy is reported to occur at any stage of PD in human patients, even in the prodromal stage when motor symptoms are not evident [10]. Based on the results of the motor function tests and pathological evaluation, our model mice correspond to the early stage of PD. Notably, 6-OHDA-lesioned mice exhibited motor symptoms despite a lesser degree of dopaminergic degeneration compared with human PD patients [49]. This may be because 6-OHDA administration induced rapid dopaminergic denervation in animals, whereas this process takes several years to decades in humans.

In the experiments of apathy-like behavior, we found that the 6-OHDA group exhibited significant impairments in novelty seeking. Notably, PD patients may exhibit low novelty-seeking behavior [50]. Although the underlying neural mechanism of novelty-seeking behavior in PD patients is largely unknown, it is assumed that the midbrain dopaminergic system plays an essential role [51]. The activity of VTA/SNc dopaminergic neurons has been reported to correlate with an increase or a decrease in novelty-seeking behavior in mice [52,53]. The present PD mouse model exhibited dopaminergic neuronal loss in the dorsal striatum and impairments in novelty seeking. Low novelty seeking, which was observed in the present study, is considered a symptom with a strong cognitive apathy component in PD. The dorsal striatum, which was lesioned in the present study, is thought to be involved in cognitive apathy [10]. Additionally, a detailed anatomical study found that the dorsolateral PFC is involved in executive processing in cooperation with the ACC in cognitive apathy. The dorsolateral PFC projects neural circuits to the dorsal striatum [12], which was the target of 6-OHDA lesioning in our PD model. Therefore, our mice may be considered a model of cognitive apathy, especially low novelty seeking. With regard to novelty seeking, recent meta-analyses revealed that in personality profiles, PD patients tend to exhibit significantly lower tendencies to explore novelty compared with healthy controls [54]. These changes in personality and motivational tendencies are known as hypodopaminergic behavioral patterns [55]. Dopamine receptor agonist administration in PD patients who performed a feedback-based probabilistic classification task showed an increase in novelty seeking [56]. Thus, lower novelty seeking may be a more universal problem among PD patients in general, with or without concomitant apathy. Few studies have examined novel exploratory behavior in animal models of PD [57,58]. Thus, the present results provide important insights into this universal problem among PD patients.

The latency to eat food was significantly longer in the 6-OHDA group in the NSFT, suggesting that these mice exhibited anhedonia. In contrast, no difference in the percentage of sucrose solution consumption was found between the sham group and 6-OHDA group in the SPT, another test of anhedonia. 

In the SPT, the score dispersion was particularly high in the sham group. This may be attributable to bias that is caused by some factors. With regard to the test duration, one rat study suggested that 1 h is insufficient because the animals drink water regardless of taste immediately after 24 h of water deprivation because of thirst [59]. Hence, we conducted a 2 h SPT, although this may have not been sufficient and led to the score dispersion. Furthermore, we imposed water deprivation according to previous studies that reported that 10–15% of mice do not drink water without first being water deprived [31]; thus, we imposed water deprivation to avoid this caveat. Nevertheless, water deprivation was shown to lead to an increase in the preference threshold for the sucrose solution and may affect the rate of development of preference and thus the overall results [60,61].

A motivational factor was added in the NSFT. In the NSFT, the desire to eat under fasting conditions conflicts with the fear of novel spaces [62]. Although the fear of novel spaces has some similarities to the OFT, no anxious behavior was suggested in the present study. However, the reduction in exploratory behavior, as suggested by the HBT results, may have influenced the NSFT in the 6-OHDA group. 

Additionally, motor impairments (i.e., the latency of action) may more greatly affect performance in the NSFT than in the ST. However, the total distance traveled and latency from the start of the test to transfer to another zonein the OFT did not significantly differ between the 6-OHDA and sham groups. This suggests that the motor effect was not considerable. However, we did not assess these parameters in the NSFT. Another possibility is that the conflicting results between the NSFT and SPT might be explained by lower motivation to eat or a decline in the awareness of one’s hunger state, but further studies are required to investigate this possibility.

Interestingly, a decrease in object/inanimate novelty seeking was suggested by the HBT results and session 1 in the three-chamber test, but social novelty seeking was maintained in the 6-OHDA group in session 2 in the three-chamber test. The possibility that VTA/SNc dopaminergic neurons also mediate social novelty has been proposed [63]. However, a recent study suggests that another midbrain region, the interpeduncular nucleus, but not the VTA, controls social novelty familiarization [62]. These findings and the present results suggest that different neural mechanisms beyond the dorsal striatum may play important roles in social novelty seeking.

As described above, the present mouse model of PD exhibited cognitive apathy-like behavior. However, emotional-affective apathy was not observed. Indeed, emotional-affective apathy is mediated by the mesocorticolimbic pathway or the reward system. The involved regions include the orbitomedial PFC, ACC, ventral striatum, ventral pallidum, and dopaminergic midbrain neurons. This system is mediated by the amygdala, hippocampus, thalamus, lateral habenular nucleus, dorsal PFC, pedunculopontine nucleus, and raphe nucleus in the brainstem [12]. In the present study, the lesion was predominantly located in the dorsal striatum and there is little direct involvement with the foci responsible for the emotional-affective apathy proposed above. Therefore, this type of apathy is less likely to be related to our model mouse. 

The present study has several limitations. One limitation of the present study was that cholinergic neurons were not manipulated or assessed in the present mouse model. Cholinergic neurons are also engaged in the nervous system to impact apathy, especially the cognitive subtype [10,12,64]. Although studies of the relationship between neuropsychiatric symptoms of PD and cholinergic dysfunction are limited, an interplay between cholinergic dysfunction and neuropsychiatric symptoms has been suggested [65]. On the other hand, some previous studies reported the compensatory upregulation of acetylcholine in PD patients [66,67]. Abnormalities in dopamine neurons are also associated with executive dysfunction [68] or cognitive dysfunction [69,70]. Therefore, we presume that dopamine loss itself may cause symptoms of apathy, regardless of changes in cholinergic function. Future studies will reveal whether the cholinergic system is involved in the appearance of apathy-like symptoms in these mice. Next, we describe limitations on the experimental method. Although the number of mice that were tested in the present study may be relatively small, recent studies of psychiatric symptoms that used the 6-OHDA lesioning model of PD in mice had group sizes of 6 to 11, reporting significant differences in the behavioral results [44,71]. When possible, it is preferable to reduce the number of animals used from an animal ethics standpoint. Even if we used desipramine, we cannot totally exclude possible toxic effects on noradrenergic neurons, which would affect PD symptoms [72]. Anxiety-like behavior was not suggested by the OFT in the present study. Other tests, such as the elevated plus maze test, would have been a good additional control for the absence of anxiety-like behavior.

## 4. Materials and Methods

### 4.1. Animals

Eight-week-old male C57BL/6J mice (21–25 g; CLEA Japan, Tokyo, Japan) were housed in our animal facility that was maintained at 23 °C ± 1 °C and 55% ± 5% relative humidity under a 12 h/12 h light/dark cycle (lights on at 8:00 AM, off at 8:00 PM). Food and water were freely available. The experimental procedures and housing conditions were approved by the Institutional Animal Care and Use Committee (Animal Experimentation Ethics Committee, Tokyo Metropolitan Institute of Medical Science; approval no. 23-009). All animals were cared for and treated humanely in accordance with our institutional animal experimentation guidelines.

### 4.2. Experimental Design

The experiments began after 2 weeks of habituation (days 1–14) to our animal facility. Three motor performance tests, including the rotarod, pole, and balance beam tests, were conducted before 6-OHDA lesioning (pretest). A total of 21 mice were assigned to sham (n = 8) and 6-OHDA lesion (n = 13) groups to ensure that there were no differences in the motor performance tests. The 6-OHDA lesioning procedure was then performed. After 3 weeks of recovery from lesioning, when the effect of 6-OHDA was expected to stabilize [73], motor performance tests were conducted again (posttest), and then seven tests that are related to apathy-like behavior were conducted. To avoid influencing each other, the behavioral experiments were conducted during different periods of time, with intervals of a few days between them, based on previous reports [48,74]. We then performed TH immunostaining. All raw data are provided in Appendix A.

### 4.3. 6-Hydroxydopamine Lesioning (Day 29)

All mice were intraperitoneally (i.p.) anesthetized with a mixture of Domitor (1.0 mg/mL; Nippon Zenyaku Kogyo, Fukushima, Japan), midazolam (5 mg/mL; Astellas Pharma, Tokyo, Japan), butorphanol (5.0 mg/mL; Meiji Seika Pharma, Tokyo, Japan), and water and mounted in a stereotaxic frame (Muromachi Kikai, Tokyo, Japan). All mice were administered 25 mg/kg desipramine (Sigma-Aldrich, St. Louis, MO, USA) that was dissolved in saline 30 min before the injection of 6-OHDA (Sigma-Aldrich) to prevent lesions of noradrenergic fibers. 6-OHDA was dissolved in saline that contained 0.02% ascorbic acid. The 6-OHDA group mice received bilateral injections of 1 μL of 6-OHDA (4 μg/μL) in the dorsolateral striatum according to the following coordinates relative to the bregma, with reference to a previous report [48]: anterior/posterior, +0.5 mm; medial/lateral, ±2.2 mm; dorsal/ventral, −3.0 mm. Control sham-lesioned mice were injected with the same volume of vehicle (0.9% saline and 0.02% ascorbic acid). After surgery, the animals were allowed to recover for 3 weeks. Immediately and the next day after surgery, all mice received fluid therapy (5% sterile glucose solution, i.p., 10 mL/kg) to prevent dehydration. Among all individual mice, one mouse from the 6-OHDA group died 13 days after lesioning. The postoperative survival rate in the 6-OHDA group was 92.3%.

### 4.4. Behavioral Assessment (Motor Performance Tests)

#### 4.4.1. Accelerating Rotarod (Days 15 and 16 for the Pretest and Days 50 and 51 for the Posttest)

The accelerating rotarod test was performed using a rotarod apparatus (MK630B, Muromachi Kikai). The apparatus consisted of a computer-controlled motor-driven single rotating rod (30 mm diameter). The rotation speed of the rod accelerated from 4 rotations per minute (rpm) to 40 rpm over the next 5 min. Mice that fell from the rotating rod were detected automatically by a pressure plate at the base of the apparatus. On day 1 of the test (training), the mice were placed on the rotarod for three trials at 15 min intervals. On day 2 (test), the mice were placed on the rotarod for three trials as described above, and the average latency to fall from the rotating rod was recorded for each individual [75].

#### 4.4.2. Pole Test (Days 18 and 19 for the Pretest and Days 53 and 54 for the Posttest)

A pole test apparatus, consisting of a wooden vertical pole (8 mm diameter, 600 mm height) with a small disk at the top to prevent mice from climbing over the pole, was placed in the mouse’s home cage with soft bedding. The mice were placed individually on top of the pole with their head oriented upward and allowed to descend to the floor. On day 1 (training), the mice underwent five trials to learn how to descend the pole. On day 2 (test), five pole descent trials were performed, and the average time required for the mice to orient themselves in a downward direction (T_turn_) and descend to the base of the pole (T_down_) were measured [76]. If the mice slipped down the pole or stopped during a trial, then the results were excluded from the analysis.

#### 4.4.3. Balance Beam Test (Days 22–24 for the Pretest and Days 57–59 for the Posttest)

A custom-built wooden bar (1 m length, 6 mm width) was placed 500 mm above a desk, with one end placed in a dark escape home cage. Testing was performed across 3 consecutive days. During the first 2 days (training), the mice were habituated to the escape cage for 2 min, then placed at the starting point individually, and trained to cross the balance beam to reach the escape cage three times. On the third day (test), the mice were placed at the starting point and allowed to cross the balance beam. Each mouse was tested three times, and the average time to cross the beam was measured [76]. If a mouse stopped in the middle of the bar for more than 2 s, then the results were excluded from the analysis.

### 4.5. Behavioral Assessment (Apathy-Related Behavioral Tests)

#### 4.5.1. Sucrose Preference Test (Days 79–81)

Anhedonia-related behavior was measured using the SPT [77,78]. After an 18 h period of water deprivation, each mouse was exposed to two bottles, one containing tap water and the other containing 1% sucrose solution. During the 2 h experimental time (9:00–11:00 AM), each mouse could drink freely from two bottles. The positions of the two bottles were exchanged at 10:00 AM (1 h after the start of the experiment). We conducted a control experiment to check for water leakage from the bottles and confirmed that leakage was negligibly small. Water consumption was measured by weighing the bottle before and after the test. Sucrose preference (expressed as a percentage) was calculated according to the following formula: sucrose solution intake/(sucrose solution + tap water intake).

#### 4.5.2. Novelty-Suppressed Feeding Test (Days 64 and 65)

The mice were food deprived 18 h before the NSFT. The mice were placed in the same corner of a novel, brightly lit plastic cage (290 mm × 460 mm × 260 mm) that all tested mouse had never experienced before the test. A single food pellet that was placed in the center for a 6 min test, and the latency to take a bite of the pellet while holding it with the forelimbs was recorded [79].

#### 4.5.3. Open-Field Test (Day 86)

The mice were placed individually in the center of a box that was constructed with a white acrylic wall (400 mm × 400 mm × 300 mm) at the bottom. The activity of the mice was observed for 5 min and automatically recorded and analyzed using animal behavior analysis software (ANY-MAZE version 6.1, Stoelting, Wood Dale, IL, USA). In the analysis, the area was divided into 16 equal squares. The number of times each mouse entered the four center square zone (25% of total area), the time that the mice were in the four center square zone, and the total distance traveled in the test were calculated [78].

#### 4.5.4. Hole-Board Test (Day 89)

The HBT was performed with an HBT apparatus (ST-1/WII, Muromachi Kikai). The apparatus consisted of a gray vinyl chloride box (400 mm × 400 mm × 300 mm) that had four holes in the floor. Each mouse was placed inside the box, and the number of times the mouse introduced its head in these holes was automatically counted by infrared sensors. The test lasted 3 min [80].

#### 4.5.5. Nest-Building Test (Day 71)

We conducted the NBT according to a previously reported protocol [81,82]. Each mouse was placed in a new cage in which the entire floor was covered with a Palsoft sheet (i.e., a sheet-type bedding material; Oriental Yeast, Tokyo, Japan) for 5 h (10:00 AM–3:00 PM). At the end of the test, each cage was photographed from two angles: top and side. Nest-building activity was assessed by visual examination of photographs of the formation of a nest according to a previously described scoring method (0, no sign of interaction or manipulation of the bedding material; 1, interaction with the nesting material is evident, but the material has not been gathered to a nest site; 2, nesting material has been gathered to form a flat nest with incomplete walls; 3, nesting material has been gathered to form a nest site and which has identifiable walls that form a “cup” or “bowl” shape; 4, bedding material has been gathered to form a nest site in the cage, and its walls do not constitute an incomplete dome structure; 5, nesting material has been gathered to form a nest site in the cage, and its walls completely enclose a hollow nest) [83]. In addition to this six-point (0–5) scale, 0.5 was added to the integers for intermediate cases.

#### 4.5.6. Splash Test (Day 68)

The ST started by spraying a 10% sucrose solution twice on the dorsal coat of the mouse. Because of the high viscosity of the sucrose solution, it dirties the mouse’s fur, which induces grooming behavior. After spraying the sucrose solution, behavior was videotaped for 5 min. The latency to the first bout of grooming behavior and total number of grooming bouts were assessed [84].

#### 4.5.7. Three-Chamber TEST (Days 73–75)

The three-chamber test was conducted using a sociability apparatus (SC-03M, Muromachi Kikai) to assess sociability and social novelty preference [43,85]. The apparatus consisted of three chambers. The dividing walls of the chamber were constructed of transparent plastic with small square openings (50 mm × 30 mm) that allowed access to each chamber (200 mm × 400 mm × 220 mm). The test mice were first placed in the middle chamber and allowed to explore the entire test chamber for 10 min (habituation). 

After that habituation, the three-chamber test consisting of two sessions of 10 min each was conducted. Immediately after the 10 min period, the test mice were placed in a clean holding cage. A small round wire cage (90 mm diameter, 185 mm height, with vertical bars 7 mm apart) was located in the left and right chambers, and a 25 mm sponge cube (novel object) or a 14-week-old male C57BL/6J mouse with no prior contact with the test mice was enclosed in each of the wire cages. Next, the test mice were returned to the middle chamber and allowed to explore for 10 min (sociability test; session 1). After the session, the 14-week-old male C57BL/6J mouse was again placed in the holding cage (familiar), and a second unfamiliar mouse (stranger) was enclosed in the other wire cage on the opposite side. The test mice were placed in the middle chamber and had a choice between the first, already investigated familiar mouse and the novel unfamiliar mouse for 10 min (social novelty preference test; session 2). These experiments were automatically recorded and analyzed using animal behavior analysis software (ANY-MAZE). In the analysis, the interaction zone was set at 30 mm around the holding cage. As an interaction time, the total time that the head of the test mouse was within the zone was counted.

### 4.6. Immunohistochemistry

After the behavioral tests, the mice were deeply anesthetized with pentobarbital, and then the transcardiac perfusion with phosphate-buffered saline (PBS) followed by 4% paraformaldehyde (PFA) was conducted, using a perfusion pressure pump. The PFA-perfusioned brain was removed and immersed in 4% PFA overnight. The solution was replaced with PBS and stored at 4 °C. Paraffin-embedded tissue sections (5 μm thick) were cut with a sliding microtome. For the pathological evaluation, two different levels of slices were configured according to a mouse brain atlas [86]: (i) CPu and NAcc as the cranial level (bregma +0.86 mm) and (ii) SNc and VTA as the caudal level (bregma −3.08 mm). Immunohistochemistry was conducted as described previously [87]. Briefly, the paraffin-embedded sections were deparaffinized, rehydrated, and immersed in distilled water. The sections were then autoclaved in 0.01 M citrate buffer (pH 6.0) for antigen activation. The sections were immersed in 0.3% hydrogen peroxide to remove endogenous peroxidase and treated with 5% normal goat serum for blocking. The sections were incubated with mouse anti-TH (1:1000; BD Transduction Laboratories, San Jose, CA, USA) diluted in PBS with 1% fetal bovine serum. Incubations with primary antibodies were performed overnight at 4 °C. The sections were then incubated for 1 h at room temperature with biotinylated secondary antibody (goat anti-mouse immunoglobulin G, 1:200; Vector Laboratories, Newark, CA, USA) diluted in PBS. Afterward, the sections were incubated with avidin peroxidase (1:100 in PBS) for 50 min at room temperature. The reaction product was visualized using diaminobenzidine (Sigma) in TBS with 0.0024% H_2_O_2_.

### 4.7. Quantitative Analysis of TH-Positive Cells

The immunostained slides were photographed using an inverted fluorescence phase-contrast microscope (BZX800, Keyence, Osaka, Japan). For assessment of the CPu and NAcc, bregma +0.86 mm levels were taken as the cephalic slice. For assessment of the SNc and VTA, bregma −3.08 mm levels were taken as the caudal slice. In each slice, a mouse brain atlas was used as a reference for the identification of areas to be evaluated [86]. All of the following steps were performed using ImageJ version 1.53e (National Institutes of Health, Bethesda, MD, USA). For the cephalic slice images, the photographs were converted to 16-bit monochrome images. Background regions were defined in slices that did not contain dopaminergic nerves (lateral and medial septal nucleus). The mean + 2 standard deviations of the contrast in the background region was used as the threshold, and the number of pixels in the target region that showed a contrast higher than the threshold was counted as positive for TH staining. For the caudal slice images, we used a cell count method for TH-positive cells. In the 6-OHDA group, a mouse with TH+ cell counts in the SNc that were >80% of the average of the Sham group was excluded from the analysis because of insufficient lesioning.

### 4.8. Statistical Analysis

The statistical analysis was performed using R version 4.0.5 software (R Core Team [2021]; R Foundation for Statistical Computing, Vienna, Austria). The results are presented as the mean ± standard error of the mean (SEM). The assumption of normality was tested for all continuous variables by evaluating the frequency distribution histogram and conducting the Shapiro–Wilk test. For comparisons of means between the sham and 6-OHDA groups, normally distributed data were analyzed using Student’s *t*-test or Welch’s *t*-test. The Mann–Whitney U test was conducted for non-normally distributed variables. For comparisons of results over time within the same group, normally distributed data were analyzed using a paired *t*-test and Wilcoxon signed-rank sum test for non-normally distributed variables. All of the tests were two-tailed. Values of *p* < 0.05 were considered statistically significant.

## 5. Conclusions

The present mouse model of PD that was induced by bilateral 6-OHDA injections in the dorsal striatum predominantly exhibited a decrease in novelty seeking as a symptom that is related to the cognitive apathy component of PD. Although object/inanimate novelty seeking decreased, social novelty seeking was maintained, suggesting the involvement of different neural circuits beyond the dorsal striatum. The present mouse model of PD may contribute to future therapeutic investigations of apathy based on its subdomains, especially in the treatment of cognitive-like apathy symptoms, including impairments in novelty seeking.

## Figures and Tables

**Figure 1 ijms-25-07993-f001:**
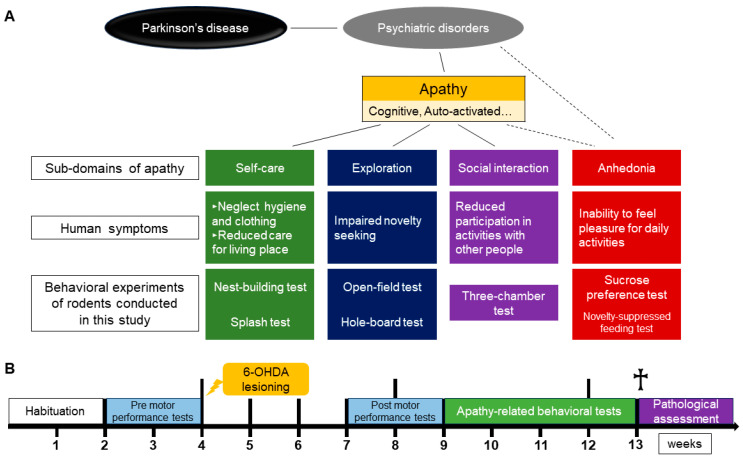
Experimental design. (**A**) Concept of this study, with correspondence between subdomains of apathy, human symptoms, and behavioral experiments in rodents. No previous studies have reported direct correspondence between human apathy subtypes and specific behavioral experiments in animal. However, a study described human clinical symptoms according to each of the apathy subtypes [12], and a mini-review proposed human clinical symptoms (not categorized by subtype) that are associated with apathy and behavioral experiments in animals to assess them [26]. We combined these reports to form the concept of the present study. Using a mouse model of PD with 6-hydroxydopamine lesions in the bilateral dorsal striatum, we examined which subtype of apathy is caused by the model and evaluated behavioral symptoms that are elicited in apathy-related behavioral experiments. (**B**) Experimental flow of the present study. After 2 weeks of habituation, three motor performance tests were conducted before 6-OHDA lesioning (pretest). A total of 21 mice were assigned to the sham (n = 8) and 6-OHDA lesion (n = 13) groups, and the 6-OHDA lesioning procedure was then performed. After 3 weeks, motor performance tests were conducted again (posttest), and then seven tests that are related to apathy-like behavior were conducted. Finally, we performed tyrosine hydroxylase (TH) immunostaining. †: euthanasia.

**Figure 2 ijms-25-07993-f002:**
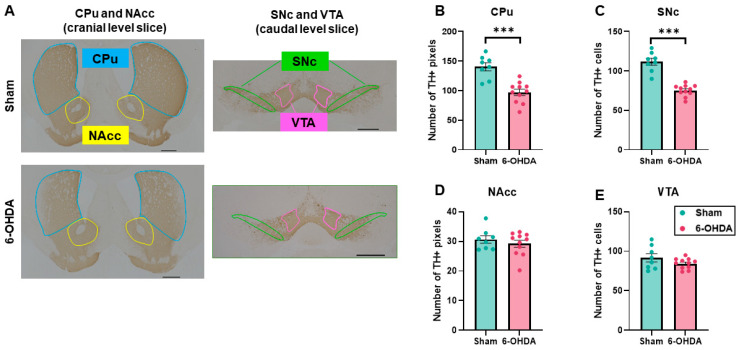
Pathological analysis. (**A**) Representative immunohistological images of tyrosine hydroxylase (TH) staining. For assessment of the caudate putamen (CPu) and nucleus accumbens (NAcc), bregma +0.86 mm levels were taken as the cephalic slice (left column). The CPu is outlined in blue, and the NAcc is outlined in yellow. For assessment of the substantia nigra compacta (SNc) and ventral tegmental area (VTA), bregma −3.08 mm levels were taken as the caudal slice (right column). The SNc is outlined in green, and the VTA is outlined in pink. In 6-hydroxydopamine (6-OHDA)-lesioned individuals (upper row), less TH staining was visible in the lateral, dorsal part of the CPu and substantia nigra compacta (SNc). Scale bar = 500 µm. (**B**,**C**) Number of TH+ pixels in the CPu and SNc. (**D**,**E**) Cell counts of TH+ cells in the NAcc and VTA. The significant loss of TH-positive pixels in the CPu and cell counts in the SNc were observed in the 6-OHDA group compared with the sham group. In the NAcc and VTA, there was no significant reduction in pixels or cell counts in the 6-OHDA group compared with the sham group. The data are expressed as the mean ± SEM with data point overlap. *** *p* < 0.001.

**Figure 3 ijms-25-07993-f003:**
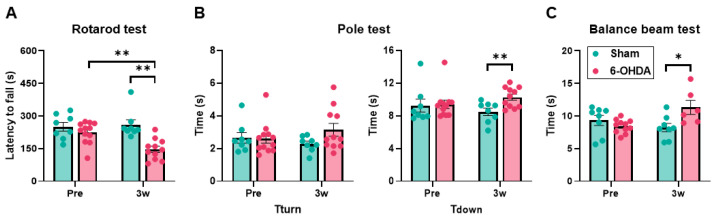
The motor performance tests. (**A**) Accelerating rotarod test. The mean latency to fall from the rotating rod was recorded in each trial. (**B**) Pole test. Two parameters were measured: time required for the mice to orient themselves in a downward direction (T_turn_) and time to descend to the base of the pole (T_down_). The average of five trials was measured for each individual. (**C**) Balance beam test. The mean time to cross the beam in three trials was measured. On the horizontal axis (x-axis) of the graph, Pre represents the results before 6-hydroxydopamine (6-OHDA) administration, and 3w represents the results 3 weeks after 6-OHDA administration. Except for T_turn_ in the pole test, all parameters showed significant deterioration in the 6-OHDA group compared with the sham group 3 weeks after 6-OHDA administration. In a pre- and post-treatment comparison within the same group, there was significant deterioration in the 6-OHDA group after treatment in the rotarod test. The data are expressed as the mean ± SEM with data point overlap. * *p* < 0.05, ** *p* < 0.01.

**Figure 4 ijms-25-07993-f004:**
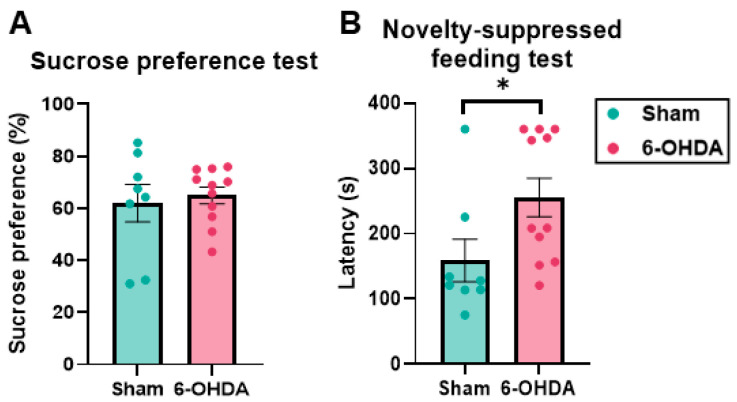
Behavioral tests of anhedonia. (**A**) Sucrose preference test (SPT). Sucrose preference (expressed as a percentage) was calculated according to the following formula: sucrose solution intake/(sucrose solution + tap water intake). (**B**) Novelty-suppressed feeding test (NSFT). The latency to take a bite of the pellet that was placed in the center while holding it with the forelimbs was recorded. In the SPT, the sucrose preference was not different between the sham and 6-OHDA groups. In the NSFT, the mean latency was significantly longer in the 6-OHDA group. The data are expressed as the mean ± SEM, with data point overlap. * *p* < 0.05.

**Figure 5 ijms-25-07993-f005:**
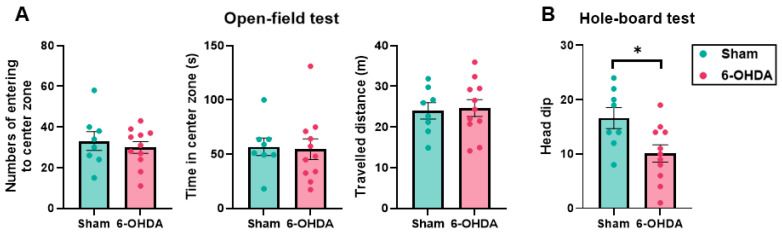
Behavioral tests of anxiety and novelty-seeking behavior. (**A**) Open-field test (OFT). (**B**) Hole-board test (HBT). The number of times each mouse entered the four center square zone (left), the duration each mouse was in the four center square zone (middle), and the total distance traveled during the test (right) were measured. None of these parameters were different between the sham and 6-OHDA groups. In the HBT, the number of times the mice introduced their head in the holes was measured and significantly decreased in the 6-OHDA group compared with the sham group. The data are expressed as the mean ± SEM with data point overlap. * *p* < 0.05.

**Figure 6 ijms-25-07993-f006:**
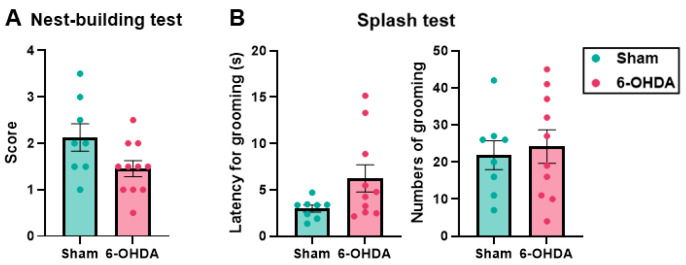
Behavioral tests of self-care behavior. (**A**) Nest-building test (NBT). (**B**) Splash test (ST). Nest-building activity was assessed by scoring the formation of a nest according to a six-point (0–5) scale, and 0.5 was added to the integers for intermediate cases. There was a nonsignificant trend toward a decrease in the nest-building score in the 6-OHDA group compared with the sham group. The ST was started by spraying a 10% sucrose solution twice on the dorsal coat of the mouse, and the following behaviors were videotaped for 5 min: latency to the first bout of grooming behavior and total number of grooming bouts. The latency to the first grooming bout increased but not significantly in the 6-OHDA group compared with the sham group. The total number of grooming behaviors was also not significantly different between the two groups. The data are expressed as the mean ± SEM with data point overlap.

**Figure 7 ijms-25-07993-f007:**
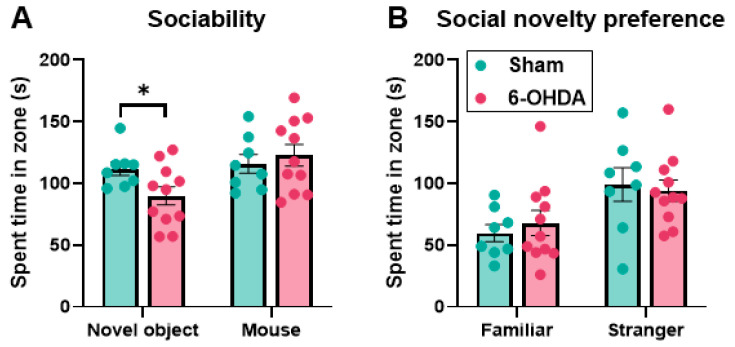
Three-chamber test. (**A**) Results of session 1, sociability test. (**B**) Results of session 2, social novelty preference. In small round wire cages in the left and right of three chambers, a 25 mm sponge cube (novel object) or a 14-week-old male C57BL/6J mouse with no prior contact with the test mouse was enclosed, respectively (session 1). The test mice were placed to the middle chamber and allowed to explore it for 10 min. To determine the interaction time, the total duration that the head of the test mouse was within the zone that was set at 30 mm around each of the cages was automatically counted using animal behavior analysis software. The interaction time for the novel object significantly decreased in the 6-OHDA group compared with the sham group, but there was no significant difference for the social object. After the session, the 14-week-old male C57BL/6J mouse was again placed in the holding cage (familiar), and a second unfamiliar mouse (stranger) was enclosed in the other wire cage on the opposite side (session 2). The rest of the procedure was conducted in the same way as mentioned above for session 1. The interaction time for either the familiar or stranger mouse did not significantly differ between the two groups. The data are expressed as the mean ± SEM with data point overlap. * *p* < 0.05.

## Data Availability

The original contributions presented in the study are included in the article/Appendix A; further inquiries can be directed to the corresponding authors.

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
