# Peer review of "Mouse Model of Parkinson’s Disease with Bilateral Dorsal Striatum Lesion with 6-Hydroxydopamine Exhibits Cognitive Apathy-like Behavior"

_ijms, 2024, doi:10.3390/ijms25147993_

Round 1
Reviewer 1 Report
Comments and Suggestions for Authors
This article examines which subtype of apathy is elicited in a mouse model of PD with 6-hydroxydopamine (6-OHDA) lesions and the behavioral symptoms that are exhibited.
The prevalence of Parkinson's disease is not necessary to be mentioned in the first paragraphs.
Limitations should be mentioned at the end and not in line 390.
Instead in the introduction authors should mention why apathy is so important and its key role in humans everyday functioning with Parkinson's disease (for a relevant recent article to discuss authors can read: doi: 10.3390/brainsci11060785).
The hypotheses should be clearly presented and supported by relevant literature. Please add.
How was the sample size of the included animals estimated? Please justify as the sample seems to be small and this may be a problem for the statistical analyses.
The order of the tests is not clear. Please specify.
Please expand more in the discussion in a more critical way on the clinical implications for humans and animals.
Comments on the Quality of English Language
Moderate English language editing.
Author Response
Dear reviewer 1
Thank you for your prompt and very valuable review.
I have replied to each of your comments below. I have also uploaded the revised manuscript. The deleted parts are in blue font and strikethrough, and the added parts are in red font.
・The prevalence of Parkinson's disease is not necessary to be mentioned in the first paragraphs.
 →Indeed, the description of the prevalence of Parkinson's disease was not relevant to the content of our study and seemed to be of little importance. Therefore, we deleted the description (Lines 41-44).
・Limitations should be mentioned at the end and not in line 390.
 →We started the limitation part on line 390 and then continued to a separate paragraph to explain the limitation of the experiment on line 399. This may have been confusing. Therefore, we clarified that there was more than one limitation (Line 413) and combined these paragraphs (Lines 413-433).
・Instead in the introduction authors should mention why apathy is so important and its key role in humans everyday functioning with Parkinson's disease (for a relevant recent article to discuss authors can read: doi: 10.3390/brainsci11060785).
 →We added a description of the significant impact of apathy on the lives of PD patients: In PD patients, apathy-associated complications are important and play a key role in daily functioning. For example, apathy can lead to a poor response to motor symptom treatment and greater healthcare spending of both patients and their caregivers, and it can negatively influence general cognitive function and financial capacity performance [6,7]. Additionally, apathy was reported to be an independent non-motor symptom that determined poor health-related quality of life in patients with early PD [8],” based on the literature that the reviewer provided (Giannouli, V.; Tsolaki, M. Is depression or apathy playing a key role in predicting financial capacity in Parkinson's disease with dementia and frontotemporal dementia? Brain Sci 2021, 11, 785, http://doi.org/ 10.3390/brainsci11060785) and others (Mele, B.; Van, S.; Holroyd-Leduc, J.; Ismail, Z.; Pringsheim, T.; Goodarzi, Z. Diagnosis, treatment and management of apathy in Parkinson's disease: a scoping review. BMJ Open 2020, 10, e037632, http://doi.org/10.1136/bmjopen-2020-037632; Choi, S. M.; Cho, S. H.; Choe, Y.; Kim, B. C. Clinical determinants of apathy and its impact on health-related quality of life in early Parkinson disease. Medicine 2023, 102, e32674, http://doi: 10.1097/MD.0000000000032674) (Lines 52-58).
・The hypotheses should be clearly presented and supported by relevant literature. Please add.
 →We described our research hypotheses on neural network disorders and apathy subtype symptoms that were expected in the present mouse model from an anatomical perspective, referring to previous reports: “Based on previous reports [10,12], we hypothesized that the corresponding neural network, including the dorsal striatum, would be disrupted and result in behavioral abnormalities that are attributable to the cognitive apathy subtype” (Lines 89-91).
・How was the sample size of the included animals estimated? Please justify as the sample seems to be small and this may be a problem for the statistical analyses.
 →We referred to recent studies in which psychiatric symptoms were examined using a 6-OHDA lesion model of Parkinson's disease in mice (Slézia, A.; Hegedüs, P.; Rusina, E.; Lengyel, K.; Solari, N.; Kaszas, A.; Balázsfi, D.; Botzanowski, B.; Acerbo, E.; Missey, F.; et al. Behavioral, neural and ultrastructural alterations in a graded-dose 6-OHDA mouse model of early-stage Parkinson’s disease. Sci Rep 2023, 13, 19478, http://doi.org/10.1038/s41598-023-46576-0; Liu, X.; Yu, H.; Chen, B.; Friedman, V.; Mu, L.; Kelly, T. J.; Ruiz-Pérez, G.; Zhao, L.; Bai, X.; Hillard, C. J.; et al. CB2 agonist GW842166x protected against 6-OHDA-induced anxiogenic- and depressive-related behaviors in mice. Biomedicines 2022, 10, 1776, http://doi: 10.3390/biomedicines10081776). In these studies, the number of mice in each group ranged from 6 to 11, although the number of allocated groups was four, and they found significant differences in the behavioral results. As mentioned by the reviewer, the number of animals in the present study was relatively small. However, this is preferable from an ethics point of view and also because it was the maximum number of animals per group with which we could conduct many behavioral experiments under the same conditions. A summary of the above was added as a limitation (Lines 425-429).
・The order of the tests is not clear. Please specify.
 →We specify the days when the behavioral experiments and drug administration were conducted in the Materials and Methods.
・Please expand more in the discussion in a more critical way on the clinical implications for humans and animals.
 →We added a description of the significant clinical relevance of impairments in novelty seeking, which is a particularly important result in our study, and we cited new references (Lines 358-365). We also included a description of the animal model of Parkinson's disease, which we believe to be of value in this study because there have been few studies of animal models of Parkinson's disease with regard to novelty seeking (Lines 365-367).
Comments on the Quality of English Language
Moderate English language editing.
 →However, please consider the fact that all the files including the revised manuscript and even the reply have been edited by a native English speaker, described in the Acknowledgments, who has extensive experience in editing medical papers.
Best regards,
Masato Okitsu.
Addictive Substance Project, Tokyo Metropolitan Institute of Medical Science, Tokyo, Japan
Department of Neurology, Tokyo Metropolitan Neurological Hospital, Tokyo, Japan
Reviewer 2 Report
Comments and Suggestions for Authors
I appreciate the authors for presenting this research article. The introduction is well-reviewed but does not provide the hypothesis. Other comments are as follows:
Materials and Methods
1. Experimental Design (Figure 1B)
a. It’s not clear what the exact time points for evaluations are, such as pre-motor performance, post-motor performance, and apathy-related behavior. Please mark the key points.
b. Cite references to justify your choice of different periods for evaluation (2-4 weeks, 7-9 weeks, and 9-13 weeks).
2. 6-Hydroxydopamine Lesioning
a. What are the inclusion criteria for successful 6-Hydroxydopamine lesioning?
Behavioral Assessment
3. Ten behavior tests are mentioned from sections 2.4.1 to 2.4.10. It would be better to separate them into motor performance tests (3 tests) and apathy-related behavior tests (7 tests).
Discussion
a. In the first section of the discussion, please mention the novel findings of the current study.
b. Lines 323-324 state, “Therefore, the present mouse model can be used to determine the apathy-like phenotype in the earlier stage of PD.”
What is the apathy phenotype in the earlier stages of PD in humans?
Which behaviors in the animal model are similar to the apathy-like phenotype in the earlier stages of PD?
Author Response
Dear reviewer 2
Thank you for your prompt and very valuable review.
I have replied to each of your comments below. I have also uploaded the revised manuscript. The deleted parts are in blue font and strikethrough, and the added parts are in red font.
Materials and Methods
- Experimental Design (Figure 1B)
・a. It’s not clear what the exact time points for evaluations are, such as pre-motor performance, post-motor performance, and apathy-related behavior. Please mark the key points.
→We specify the days when the behavioral experiments and drug administration were conducted in the Materials and Methods.
・b. Cite references to justify your choice of different periods for evaluation (2-4 weeks, 7-9 weeks, and 9-13 weeks).
 →The reasons for the 3-week interval between the motor function tests before and after 6-OHDA administration (Line 450) and the fact that the behavioral tests were conducted in separate time periods with an interval of several days to avoid the influence of each behavioral test on each other were described with references to previous reports (Lines 452-454).
- 6-Hydroxydopamine Lesioning
・a. What are the inclusion criteria for successful 6-Hydroxydopamine lesioning?
 →We added specific exclusion criteria based on pathological evaluation in the Materials and Methods (Lines 611-613).
Behavioral Assessment
・3. Ten behavior tests are mentioned from sections 2.4.1 to 2.4.10. It would be better to separate them into motor performance tests (3 tests) and apathy-related behavior tests (7 tests).
 →As indicated by the reviewer, this would make it easier for readers to understand. We divided them into separate sections (Lines 473, 503).
Discussion
・a. In the first section of the discussion, please mention the novel findings of the current study.
 →We described the novelty and value of this study at the end of the first section of the Discussion: “The present study is novel and valuable in that animal behavioral experiments showed reduced novelty seeking seems to be a symptom with a strong cognitive apathy component among the apathy symptoms of PD. Furthermore, it is also interesting to note that while object/inanimate novelty seeking decreased, social novelty was maintained.” (Lines 320-323).
・b. Lines 323-324 state, “Therefore, the present mouse model can be used to determine the apathy-like phenotype in the earlier stage of PD.”
What is the apathy phenotype in the earlier stages of PD in humans?
Which behaviors in the animal model are similar to the apathy-like phenotype in the earlier stages of PD?
 →We agree that the expression mentioned by the reviewer (Line 335-336) was misleading, considering the connection between the preceding and subsequent paragraphs. We wanted to emphasize that our mouse model corresponds to the early stage of Parkinson's disease, based on the results of the motor function tests and pathological evaluation. Therefore, we revised the sentence: “Based on the results of the motor function tests and pathological evaluation, our model mice correspond to the early stage of PD” (Lines 337-338).
Best regards,
Masato Okitsu.
Addictive Substance Project, Tokyo Metropolitan Institute of Medical Science, Tokyo, Japan
Department of Neurology, Tokyo Metropolitan Neurological Hospital, Tokyo, Japan